# Amelioration of Sensorineural Hearing Loss through Regulation of *Trpv1*, *Cacna1h*, and *Ngf* Gene Expression by a Combination of Cuscutae Semen and Rehmanniae Radix Preparata

**DOI:** 10.3390/nu15071773

**Published:** 2023-04-05

**Authors:** Bin Na Hong, Sung Woo Shin, Youn Hee Nam, Ji Heon Shim, Na Woo Kim, Min Cheol Kim, Wanlapa Nuankaew, Jong Hwan Kwak, Tong Ho Kang

**Affiliations:** 1Department of Oriental Medicine Biotechnology, Graduate School of Biotechnology, Kyung Hee University, Global Campus, Yongin 17104, Gyeonggi-do, Republic of Korea; habina22@hanmail.net (B.N.H.); 01073205620@khu.ac.kr (S.W.S.); 01030084217@hanmail.net (Y.H.N.); jee1015235@gmail.com (J.H.S.); nawoonifty@khu.ac.kr (N.W.K.); kminchel1213@gmail.com (M.C.K.); wanlapa.nuankaew@gmail.com (W.N.); 2Invivotec Co., Ltd., Seongnam 13449, Gyeonggi-do, Republic of Korea; 3School of Pharmacy, Sungkyunkwan University, Suwon 16419, Gyeonggi-do, Republic of Korea

**Keywords:** sensorineural hearing loss, TS, Cuscutae Semen, Rehmanniae Radix Preparata

## Abstract

Sensorineural hearing loss (SNHL) is a common condition that results from the loss of function of hair cells, which are responsible for converting sound into electrical signals within the cochlea and auditory nerve. Despite the prevalence of SNHL, a universally effective treatment has yet to be approved. To address this absence, the present study aimed to investigate the potential therapeutic effects of TS, a combination of Cuscutae Semen and Rehmanniae Radix Preparata. To this end, both in vitro and in vivo experiments were performed to evaluate the efficacy of TS with respect to SNHL. The results showed that TS was able to protect against ototoxic neomycin-induced damage in both HEI-OC1 cells and otic hair cells in zebrafish. Furthermore, in images obtained using scanning electron microscopy (SEM), an increase in the number of kinocilia, which was prompted by the TS treatment, was observed in the zebrafish larvae. In a noise-induced hearing loss (NIHL) mouse model, TS improved hearing thresholds as determined by the auditory brainstem response (ABR) test. Additionally, TS was found to regulate several genes related to hearing loss, including *Trpv1*, *Cacna1h*, and *Ngf*, as determined by quantitative real-time polymerase chain reaction (RT-PCR) analysis. In conclusion, the findings of this study suggest that TS holds promise as a potential treatment for sensorineural hearing loss. Further research is necessary to confirm these results and evaluate the safety and efficacy of TS in a clinical setting.

## 1. Introduction

Hearing loss is a prevalent condition that is characterized by difficulties in perceiving sounds or words. The classification of hearing loss is based on the cause of its occurrence, with conductive hearing loss and sensorineural hearing loss being the two main categories [1]. Sensorineural hearing loss (SNHL) is often associated with aging, exposure to noise, and ototoxic drugs that damage the hair cells in the cochlea. In recent years, the prevalence of hearing loss has increased due to factors such as the widespread use of earphones, stress, and sleep deprivation [2]. Moreover, the side effects of COVID-19 have contributed to the emergence of hearing loss as a social problem [3,4]. Sensorineural hearing loss is caused by multiple pathogenic mechanisms [5]. Natural products, containing multi-target functional components, have been proposed as potential alternatives to drugs that only have a limited number of targets [6]. Additionally, natural products are generally less toxic than chemical drugs and have a proven clinical efficacy and safety record and a long history of use [7]. These advantages reduce the time and cost of developing new drugs or functional foods and promote the long-term use of natural products.

Cuscutae Semen and Rehmanniae Radix Preparata are traditionally used in Korean medicine to treat or prevent diseases. Cuscutae Semen refers to the dried ripe seeds of *Cuscuta chinensis* Lam. [8], which contain various flavonoids such as quercetin and kaempferol [9]. Rehmanniae Radix Preparata refers to the processed roots of *Rehmannia glutinosa* Liboschitz ex Steudel, which undergoes a steaming and drying procedure that is repeated nine times [10]. In Korean medicine, Rehmanniae Radix is traditionally used in three forms: raw, dried, and processed. The processing of Rehmanniae Radix involves nine repetitions of a steaming and drying procedure, which is known to increase the availability of the major compound 5-HMF (5-hydroxymethyl-2-furaldehyde) [11]. It has been reported that Rehmanniae Radix Preparata has antioxidant effects that protect mouse auditory cells [12,13]. Traditional Korean medical books such as *Donguibogam* (1613) and *Bangyakhappyeon* (1884) have recorded the use of the combination of Cuscutae Semen and Rehmanniae Radix Preparata to treat the symptom of ringing in the ear, which is the perception of sound in the absence of an external sound source. Hearing loss and tinnitus are closely related, and most patients with hearing loss are reported to experience tinnitus [14,15]. Therefore, this study was conducted to confirm whether the combination of Cuscutae Semen and Rehmanniae Radix Preparata could be used to effectively treat hearing loss. In addition, in Korean, Cuscutae Semen is pronounced “to-sa-ja” and Rehmanniae Radix Preparata is pronounced “suk-ji-hwang”. Therefore, the combination of Cuscutae Semen and Rehmanniae Radix Preparate was termed “TS” according to the first letters of each word.

Recent studies have linked the genes *Trpv1*, *Cacna1h*, and *Ngf* with sensorineural hearing loss. The Transient Receptor Potential Vanilloid 1 (TRPV1) channel is a cation channel activated by heat and other physical and chemical stimuli [16]. This channel is present in the organ of Corti and mediates the cellular uptake of ototoxic aminoglycosides that increase the generation of reactive oxygen species [17,18]. In addition, the activation of TRPV1 by noise and ototoxic drugs in the inner ear leads to depolarization of the membrane and calcium influx, resulting in hearing loss [19,20,21,22]. Calcium (Ca^2+^) is a critical ion involved in regulating various cellular activities. The intracellular homeostasis of calcium ions is essential for cell survival and is altered by noise and ototoxic drugs [23,24,25]. An excessive level of intracellular Ca^2+^ can activate a series of enzymes that cause cellular damage and apoptosis [26]. The intracellular concentration of calcium is regulated by voltage-gated calcium channels (VGCC), including L, N, P/Q, R, and T-type calcium channels [27]. Among them, the T-type Ca^2+^ channels encoded by the Cacna1h gene have been reported to play a crucial role in auditory perception and information processing within the inner ear and brainstem [28]. In addition, TRPV1 activation inhibits the T-type calcium channel in rat sensory neurons [29]. In the auditory pathway, the spiral ganglion neuron (SGN) is the first afferent neuron [30], and nerve growth factor (NGF) plays a crucial role in the growth, survival, and differentiation of acoustic ganglion cells [31,32]. Loss of cochlear neurons often occurs as a result of hair cell loss, which is thought to be due to decreased neurotrophin exposure and can progress slowly, particularly in humans [33]. Patients with sensorineural hearing loss have been found to have low levels of NGF in their serum [34]. In addition, an increase in NGF has been shown to protect against ototoxic, drug-induced hair cell damage and improve hearing as measured by the Auditory Brainstem Response (ABR) test [35,36]. Therefore, the current study was conducted to confirm the effects of TS on sensorineural hearing loss by regulating the gene expression of *Trpv1*, *Cacna1h*, and *Ngf*.

In this study, the potential efficacy of TS, a combination of Cuscutae Semen and Rehmanniae Radix Preparata, with respect to treating sensorineural hearing loss was explored using cell, zebrafish, and mouse models. The results obtained provide evidence supporting the hypothesis that TS may have a positive impact on sensorineural hearing loss. Further research is needed to fully establish the efficacy of TS and determine the underlying mechanisms of its actions.

## 2. Materials and Methods

### 2.1. Preparation of TS

Cuscutae Semen and Rehmanniae Radix Preparata was mixed at a ratio of 2:1 and extracted with 70% EtOH. The combination ratio was determined using the MTT assay in HEI-OC1 cells, and the corresponding result can be seen in Appendix A. The Lot number is IVT02. TS’s active ingredients were identified, including hyperoside, quercitrin, and kaempferol. A 70% ethanol extract was prepared by heating the mixture at 80 °C for 2 h. The extract was filtered and concentrated using a vacuum evaporator at a temperature of 55 ± 5 °C. The concentrated extract was mixed with dextrin, corresponding to 30% of the amount of the concentrate, and then powdered using a spray dryer.

### 2.2. House Ear Institute-Organ of Corti 1 (HEI-OC1) Cells

The HEI-OC1 cell line was cultured in high-glucose Dulbecco’s Eagle’s medium (DMEM, Sigma-Aldrich Co., St. Louis, MO, USA) supplemented with 10% fetal bovine serum (FBS, WELGENE Inc., Gyeongsangbuk-do, Republic of Korea) and 50 U/mL INF-γ (Peprotech Inc., Cranbury, NJ, USA) in accordance with previous protocols [37]. The cells were seeded at a density of 1 × 10^4^ cells/well in 96-well flat bottom plates for MTT assay or 2 × 10^6^ cells/well in 6-well cell culture plates for RT-PCR. Incubation was carried out overnight for attachment. The cells were then pre-treated with TS at a final concentration of 1 μg/mL for 1 h, followed by cotreatment with 10 mM neomycin for an additional 24 h.

### 2.3. MTT Assay

Cellular viability was determined using the MTT (3-(4,5-dimethylthiazol-2-yl)-2,5-diphenyltetrazolium bromide) assay. The HEI-OC1 cells were exposed to 0.5 mg/mL of MTT solution (Duchefa Biochemie, Amsterdam, The Netherlands) for 4 h, and the resulting formazan crystals were solubilized with 100 μL of dimethyl sulfoxide (DMSO). The absorbance was measured using a 96-well microplate reader (Synergy HT, BioTek Instruments, Winooski, VT, USA) at 570 and 630 nm. The average optical density (OD) in the control cells was taken as 100% viability.

### 2.4. Zebrafish

The adult wild-type zebrafish (*Danio rerio*) were housed in an S type (1500 (W) × 400 (D) × 2050 (H) mm) zebrafish system provided by Genomic Design Bioengineering Co., Daejeon, Republic of Korea. The water temperature in the system was maintained at a constant 28 °C. A group comprising a ratio of three males to four females was placed overnight in a spawning box, and the eggs were collected at 3 h post-fertilization. The eggs were then incubated in Petri dishes with a solution containing 0.03% sea salt (Sigma-Aldrich Co., St. Louis, MO, USA). The embryos were kept in an incubator with a 14 h light/10 h dark cycle at a temperature of 28.5 ± 0.5 °C until 6 days post-fertilization (dpf), at which point the experiments were performed. All experimental procedures carried out on zebrafish were in accordance with standard zebrafish protocols and were approved by the Animal Care and Use Committee of Kyung Hee University (KHUASP(SE)-15-10).

### 2.5. Neomycin-Induced Ototoxicity in Zebrafish

The zebrafish larvae at 6 dpf in a 96-well plate were exposed to 100 μL of 2 μM neomycin sulfate (MB Cell Co., Seoul, Republic of Korea) for 1 h to induce ototoxicity. After inducing damage to the hair cells of the zebrafish, the neomycin solution was completely extracted, and the larvae were rinsed with 0.03% sea salt solution. To evaluate the effect of TS on the recovery of the otic hair cells, the TS group was exposed to 1 μg/mL of TS for 6 h in an incubator maintained at 28 °C. After incubation, the zebrafish were rinsed with 0.03% sea salt solution and stained with 0.1% YO-PRO-1 (Thermo Fisher Scientific Inc., Gainesville, FL, USA) for 30 min, after which they were anesthetized with 0.04% tricaine (Sigma Chemical Co., St. Louis, MO, USA). The otic hair cells were counted after visualization under a fluorescence microscope (Olympus 1 × 70; Olympus Co., Tokyo, Japan). All image analyses were conducted using Focus Lite software (Focus Co., Daejeon, Republic of Korea).

### 2.6. Scanning Electron Microscopy (SEM)

The morphological differences of cilia in hair cells were assessed by means of Scanning Electron Microscopy (SEM). The 6 dpf zebrafish larvae were fixed in Phosphate-Buffered Saline (PBS, pH 7.2) containing 2.5% glutaraldehyde at 4 °C for 12 h. The fixed zebrafish were then washed three times, each for 5 min, with distilled water and underwent serial dehydration in tert-butanol mixed with ethanol (25%, 50%, 75%, and 100%) for 10 min. Following dehydration, the specimens were dried via critical point drying and coated twice with platinum using an ion sputter coater (PS-1200; Tescan, Brno, Czech Republic). SEM imaging was conducted at 10 kV using a supra55 device (Zeiss, Jena, Germany).

### 2.7. Mouse

Male Institute of Cancer Research (ICR) mice of 6 weeks of age were obtained from Orient Bio, Inc. (Seongnam, Republic of Korea), and housed under controlled conditions with a 12 h light/dark cycle at a temperature of 23.0 ± 2.0 °C and humidity of 50.0 ± 5.0%. The mice had unrestricted access to food and water. After a week of acclimation, the hearing threshold of the mice was evaluated using an auditory brainstem response (ABR) test to select mice with normal hearing ability (≤25 dB). All experimental procedures using mice were approved by the Animal Care and Use Committee of Kyung Hee University (KHUASP-15-17).

### 2.8. Auditory Brainstem Response (ABR)

Hearing loss was induced in the mice by noise exposure at 115 dB sound pressure level (SPL) for 1 h in a chamber with dimensions of 250 × 560 × 550 cm^3^ at a temperature of 20–23 °C. The stability of the stimulus was ensured by measuring the sound level at the center of the chamber using a Sound Level Meter (SL-5868P; Hanada Technology, Wenzhou, China). Twenty-four hours after noise exposure, the mice in TS group (*n* = 10) were orally administered 100 mg/kg of TS in distilled water once daily from days 1 to 7. Hearing thresholds were measured using clicks and 8, 16 kHz tone bursts on days 1, 4, and 7 post-noise exposure. The auditory brainstem response (ABR) test was performed after inducing anesthesia through intramuscular injection of xylazine (Bayer, Barmen, Germany), ketamine (Yuhan Corporation, Seoul, Republic of Korea), and a saline solution (JW Pharmaceutical, Seoul, Republic of Korea) in a 1.1:4:4.9 ratio, respectively. To protect against hypothermia, the body temperature of the experimental mice was maintained at 37 ± 1 °C during the ABR test. The ABR test was performed in an electrically and acoustically shielded sound attenuation booth (TCA-500D; SonTek, Paju, Republic of Korea). Stimuli were transmitted through earphones (Etymotic ER-EA, Elk Grove Village, Illinois, USA), and needle electrodes were located at the vertex of the skull, the post-auricular region, and on the back to record electrophysiological responses. Impedance was set below 5 kΩ and hearing thresholds were evaluated by gradually lowering the intensity by 5 dB near the threshold, with the lowest intensity that elicited a response used as the threshold. Physiological filters were set to pass electrical activity between 100 and 3000 Hz.

### 2.9. Extraction of Total RNA

Total RNA extraction from HEI-OC1 cells was performed using TRIzol reagent (Invitrogen, Carlsbad, CA, USA). The cells were homogenized in 500 μL of TRIzol solution and incubated at room temperature for 5 min to dissolve cellular components. To separate the total RNA, 100 μL of chloroform (Sigma-Aldrich) was added to the mixture and centrifuged at 10,000 rpm for 15 min at 4 °C after a 5 min incubation at room temperature. The supernatant was transferred to a new 1.5 mL microfuge tube and 250 μL of isopropanol (Samchun Chemical Co., Ltd., Seoul, Republic of Korea) was added, which was followed by a 10 min incubation at room temperature to precipitate the RNA. The mixture was then centrifuged for 10 min at 10,000 rpm at 4 °C and the supernatant was discarded. The RNA pellet was purified by adding 500 μL of 75% ethanol; then, it was centrifuged for 5 min at 8000 rpm at 4 °C and the supernatant was discarded. The RNA pellet was dried for 5–10 min at 55 °C and resuspended in DEPC water (Invitrogen, Carlsbad, CA, USA). The quality and quantity of the total RNA were analyzed using NanoDrop 2000 (Thermo Fisher Scientific Inc., Ganseville, FL, USA), Qubit (Invitrogen, Carlsbad, CA, USA), and TapeStation (Agilent Technologies, Palo Alto, CA, USA).

### 2.10. Quantitative RT-PCR

Complementary DNA (cDNA) was synthesized using 1 μg of total RNA and ReverAid First Strand cDNA Synthesis Kit (Thermo Fisher Scientific Korea Ltd., Seoul, Republic of Korea) according to the manufacturer’s instructions. mRNA expression was analyzed by real-time polymerase chain reaction (RT-PCR) with SYBR Green Master mix (Applied Biosystems, Thermo Fisher Scientific Korea Ltd., Seoul, Republic of Korea). The primer sequences used for RT-PCR are listed in Table 1. RT-PCR was carried out under the following conditions: one cycle at 95 °C for 5 min, followed by 45 cycles of 95 °C for 15 s, 60 °C for 15 s, and 72 °C for 20 s; an additional cycle at 72 °C for 20 s; and then an increase in temperature to 95 °C at 0.1 °C/s to confirm the specificity of each PCR reaction. The RT-PCR was run in triplicates using a total reaction volume of 10 μL on a Rotor Gene 6000 (Qiagen, Hilden, Germany). The expression of the housekeeping gene *β-actin* was used as an internal control for normalization purposes. The relative gene expression levels were calculated using the 2^−ΔΔCt^ method [38].

### 2.11. Statistical Analysis

Data analysis was performed using the GraphPad Prism software version 5 (GraphPad Software, Inc., San Diego, CA, USA). The results are expressed as the mean ± standard error of the mean (SEM). Statistical significance was evaluated using a one-way analysis of variance (ANOVA) test followed by Tukey’s multiple comparison test. Hearing Threshold analysis was performed by two-way ANOVA test followed by Bonferroni post hoc test. The paired t test was employed to evaluate data when needed. A significance level of *p* < 0.05 was established as the criterion for statistical significance.

## 3. Results

### 3.1. Effect of TS on Cell Viability in Neomycin-Treated HEI-OC1 Cells

The cell viability of HEI-OC1 cells was assessed using the 3-(4,5-Dimethylthiazol-2-yl)-2,5-diphenyltetrazolium bromide (MTT) assay. The effect of neomycin on cell viability was previously reported in a study by Nam et al. [39], which demonstrated a dose-dependent decrease in the cell viability of HEI-OC1 cells upon treatment with neomycin. In the current study, the neomycin-treated group (NM) was exposed to 10 mM of neomycin and showed a significantly lower level of cell viability compared to the normal group (NOR) (*p* < 0.001). However, the treatment with 1 μg/mL TS was able to significantly increase the cell viability of the HEI-OC1 cells compared to the NM group (*p* < 0.001), as illustrated in Figure 1.

### 3.2. Effect of TS on Otic Hair Cells in Neomycin-Treated Zebrafish

The exposure of zebrafish larvae to neomycin for 1 h resulted in significant ototoxicity, as evidenced by a decreased number of otic hair cells in the neomycin-treated group (NM) compared to the normal group (NOR), with a *p*-value of less than 0.001. Treatment with 1 μg/mL of TS led to the recovery of the number of hair cells in the NM group, for which there was a significant increase (*p* < 0.01), as illustrated in Figure 2.

### 3.3. Effect of TS on Cilia Bundle in Neomycin-Treated Zebrafish

Scanning electron microscopy (SEM) was performed to observe the morphological differences of cilia in each group. Figure 3 shows the SEM images, which revealed the presence of stereocilia and kinocilia. A decrease in the number of kinocilia was observed after the neomycin treatment (NM). However, the treatment with 1 μg/mL TS recovered the decreased number of kinocilia.

### 3.4. Effect of TS on Hearing Threshold in Mice with Noise-Induced Hearing Loss

Mice with abnormal hearing were excluded from the experiment, and the hearing threshold was determined as the lowest intensity required to generate an identifiable ABR waveform. The noise-induced hearing loss (NIHL) group was compared to the pre-tested (PRE), which was pre-tested before exposure to noise. As shown in Figure 4, the results indicated that the NIHL group had a significantly higher hearing threshold compared to the PRE group (*p* < 0.001). These results showed significant effects of the TS treatments on decreasing threshold of the click, 8 kHz, and 16 kHz testing stimuli (Two-way ANOVA *p* < 0.0001 on treatment effect in all three ABR test). Furthermore, the TS group showed a significantly decreased hearing threshold with respect to the click, 8 kHz, and 16 kHz stimuli (*p* < 0.001 in all three ABR test, paired *t* test) on 4 day (4D). On day 7 (7D), the TS group showed a significantly decreased hearing threshold with respect to the click (*p* < 0.01, paired *t* test), 8 kHz (*p* < 0.001, paired *t* test), and 16 kHz (*p* < 0.001, paired *t* test) stimuli.

### 3.5. Effect of TS on Gene-Expression-Related Hearing Loss in Neomycin-Treated HEI-OC1 Cells

The genes *Trpv1*, *Ngf*, and *Cacna1h*, which have been reported to be closely related to hearing loss, were selected for analysis. As shown in Figure 5, the results showed that the expression of *Trpv1* was significantly upregulated (*p* < 0.001), while the expressions levels of *Ngf* and *Cacna1h* were significantly downregulated (*p* < 0.001 and *p* < 0.01, respectively) in the neomycin-treated group (NM) compared to the normal group (NOR). However, in the TS-treated group (NM + TS), the expression of *Trpv1* was significantly downregulated (*p* < 0.05), and the expression levels of *Ngf* and *Cacna1h* were significantly upregulated (*p* < 0.05 and *p* < 0.01, respectively).

## 4. Discussion

The process of transmitting sound within the cochlea is dependent on the hair cells in the organ of Corti [40]. The cochlear hair cells consist of three rows of outer hair cells (OHCs) and one row of inner hair cells (IHCs) [41]. Sound vibrations are transmitted to the brain as electromechanical signals by moving the stereocilia on these hair cells [42]. Sensorineural hearing loss occurs as a result of damage to these hair cells via noise exposure, ototoxic drugs, and aging. Eventually, patients with sensorineural hearing loss experience difficulty in terms of sound perception and poor quality of life [43]. Currently, the number of hearing loss patients is increasing globally, yet there are limited treatments available, including medication, hearing aids, and surgical procedures such as cochlear implantation. Thus, studying the treatment of sensorineural hearing loss is critical for enhancing the quality of life for individuals with this condition. In this study, a combination of Cuscutae Semen and Rehmanniae Radix Preparata, termed TS, was developed and its effect on sensorineural hearing loss (SNHL) was investigated both in vitro and in vivo.

HEI-OC1 cells, also known as House Ear Institute-Organ of Corti 1 cells, constitute a cell line that was derived from the auditory organ of a transgenic mouse [37,44]. These cells have been widely used as in vitro models to study the physiology and pathology of the auditory system, particularly in the context of hearing loss. They have been found to express many of the same proteins and genes as the sensory cells in the organ of Corti [45]. This renders them a valuable tool for studying the molecular and cellular mechanisms underlying hearing loss and other auditory disorders. Neomycin is an aminoglycoside antibiotic that causes ototoxicity by generating toxic reactive oxygen species (ROS) in the inner ear hair cells [46,47] and induces cell death in HEI-OC1 cells [48]. To investigate its otoprotective effect, we confirmed the levels of cell viability afforded by the TS treatment applied to the neomycin-treated HEI-OC1 cells. Firstly, we treated HEI-OC1 cells with neomycin for 24 h and confirmed a significant decrease in cell viability (*p* < 0.001). The treatment of TS alone in HEI-OC1 cells did not result in any significant changes in cell viability. However, the decreased cell viability observed in the neomycin-treated HEI-OC1 cells was significantly recovered when co-treated with TS (*p* < 0.001). This result may suggest that TS has an otoprotective effect against neomycin-induced ototoxicity in HEI-OC1 cells.

In recent years, the use of zebrafish as a model organism in the study of human diseases has become increasingly important [49]. The zebrafish genome has been extensively studied and has a homologous gene for 84% of the genes known to be associated with human diseases [50]. Hair cells in zebrafish are homologous to those found in the human inner ear [51] and can be found in both the inner ear and lateral line systems of zebrafish. The lateral line system develops rapidly, and its sensory organs mature within the first week of a zebrafish’s larval development [52]. Zebrafish have been demonstrated to possess hair cells with morphological and functional properties that are similar to those found in mammals [53,54]. Compared to mammals, whose inner ears are surrounded by bones, the zebrafish model allows for the direct visualization of the hair cells in living organisms using fluorescent staining or markers under a microscope. Neomycin is known to cause hair cell death in zebrafish [55,56,57,58], much like it does in HEI-OC1 cells. Various natural products have been reported to increase the number of hair cells in neomycin-damaged zebrafish larvae [59,60]. Therefore, the effect of TS treatment on the recovery of neomycin-induced hair cell loss was evaluated in zebrafish larvae. As a result, a significant decrease in the number of hair cells of the zebrafish exposed to neomycin for one hour was observed (*p* < 0.001). Neomycin-induced damage to hair cells in zebrafish is reported to be recovered over time [61]. However, zebrafish treated with TS after neomycin-induced damage showed a significant increase in hair cell numbers (*p* < 0.001) compared to zebrafish treated with 0.03% sea salt solution for the same duration. The ability of TS was additionally verified by visualizing the apical surface of the neuromast organ of the lateral line system using scanning electron microscopy. In SEM images, stereocilia and kinocilia can be observed, and it was observed that the number of kinocilia decreased after the treatment with neomycin but increased after the treatment with TS. Although kinocilia are not directly involved in the mechanical–electrical transduction process of the auditory system, they retain certain motile characteristics that are responsible for the response of hair cells to sound stimuli [62]. Therefore, this result may suggest that TS can restore hair cells damaged by neomycin in zebrafish larvae.

Auditory brainstem response (ABR) testing is a diagnostic tool that records the electrical activity in the auditory nerve and brainstem in response to sound stimulation [63]. Mouse models have been commonly used for ABR testing because their cochlear anatomy and physiology are similar to those of humans [64] and due to the genetic similarities between the two species [65]. The hearing range of mice extends beyond the limit of human hearing, with a frequency range of approximately 1 to 100 kHz [66]. Validation studies were conducted with respect to ABR testing using ICR mice, and the results were consistent with the findings of this study [67]. The validation results demonstrated that the typical threshold for ABR testing in ICR mice with normal hearing was 20 dB or lower. The auditory threshold refers to the lowest intensity of sound that can be detected by the auditory system. However, noise exposure has been shown to cause damage to the sensory hair cells in the organ of Corti [68], resulting in significant shifts in threshold in ABR tests in response to clicks and tone burst stimuli at 8, 16, and 32 kHz [69]. This is consistent with the findings of the present study, which demonstrated a significant increase in threshold levels when measured using clicks and 8 and 16 kHz stimuli after noise exposure (*p* < 0.001). To investigate the effect of TS on the increased threshold levels following noise exposure, ABR tests using clicks and 8 and 16 kHz stimuli were conducted on ICR mice on the first, fourth, and seventh day after noise exposure. The mice were orally administered TS at a dose of 100 mg/kg once a day for seven consecutive days, starting from the day following noise exposure. The results showed that on the fourth day, TS significantly reduced the threshold levels in the mice exposed to noise compared to the NIHL group, which only received distilled water after noise exposure, in terms of the click and 8 and 16 kHz stimuli, and this trend continued to a similar degree on the seventh day. A decrease in the threshold levels on ABR tests indicates an improvement in hearing sensitivity. In this study, the significant decrease in the threshold levels in the mice administered TS after noise exposure suggests that TS may exhibit a protective effect against NIHL.

Sensorineural hearing loss can be caused by a variety of factors and mechanisms. Understanding the underlying mechanisms of sensorineural hearing loss is important for developing effective prevention and treatment strategies. Accordingly, recent studies have demonstrated the association between *Trpv1*, *Cacna1h*, and *Ngf* genes and the development of sensorineural hearing loss. Therefore, HEI-OC1 cells were used for an RT-PCR analysis to confirm the gene expression changes associated with SNHL via treatment with TS. As a result, no significant differences in the expression of the three genes were observed when the HEI-OC1 cells were treated with TS alone. However, neomycin treatment altered the expression of genes, upregulating the *Trpv1* gene and downregulating the *Cacna1h* and *Ngf* genes. This could be attributed to damage to the hair cells in the cochlea or the auditory nerve itself in the auditory pathway. These changes in gene expression were reversed upon treatment with TS. These findings suggest that TS may have a regulatory effect on the expression of genes involved in neomycin-induced ototoxicity in HEI-OC1 cells. Further studies are needed to elucidate the underlying mechanisms of this regulatory effect and explore the potential therapeutic applications of TS for the prevention or treatment of sensorineural hearing loss.

## 5. Conclusions

In conclusion, the current study provides evidence for the potential of TS, a combination of Cuscutae Semen and Rehmanniae Radix Preparata, as a therapeutic approach for the treatment of sensorineural hearing loss. The results of the in vitro and in vivo experiments suggest that TS exerts protective and ameliorative effects against sensorineural hearing loss. Further research is needed to fully understand TS’s underlying mechanisms of action with respect to sensorineural hearing loss. Nevertheless, these findings suggest that TS may hold promise as a functional food and medicine for the treatment of sensorineural hearing loss.

## Figures and Tables

**Figure 1 nutrients-15-01773-f001:**
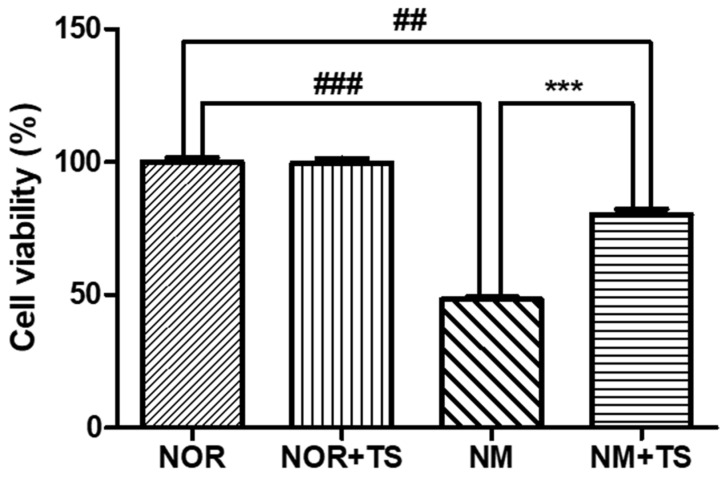
Comparison of cell viability in neomycin-treated HEI-OC1 cells with and without TS treatment. Data are presented as means ± SEM of three independent experiments conducted in triplicates. ## *p* < 0.01 (NOR vs. NM + TS). ### *p* < 0.001 (NOR vs. NM). *** *p* < 0.001 (NM vs. NM + TS).

**Figure 2 nutrients-15-01773-f002:**
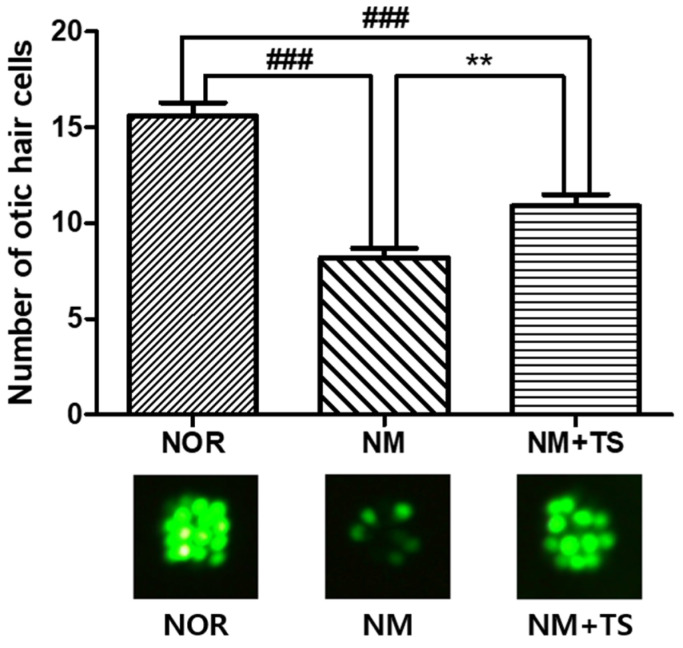
Comparison of Otic hair cells in neomycin-treated zebrafish with and without TS treatment. Data are presented as means ± SEM. ### *p* < 0.001 (NOR vs. NM and NOR vs. NM + TS). ** *p* < 0.01 (NM vs. NM + TS). N = 10 per group.

**Figure 3 nutrients-15-01773-f003:**
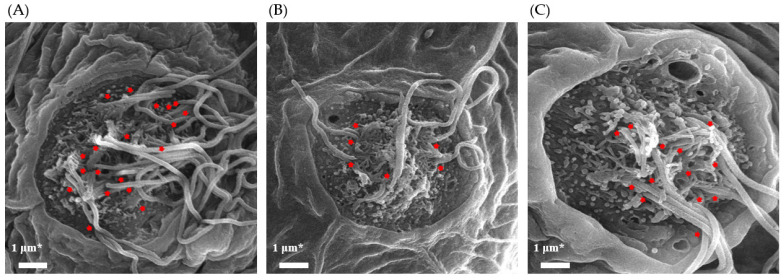
SEM images of neuromasts in zebrafish larvae: (**A**) normal group (NOR); (**B**) neomycin-treated group (NM) (**C**) TS-treated group (NM + TS). Identifiable kinocilia are indicated with red arrows. * Scale bar = 1 μm.

**Figure 4 nutrients-15-01773-f004:**
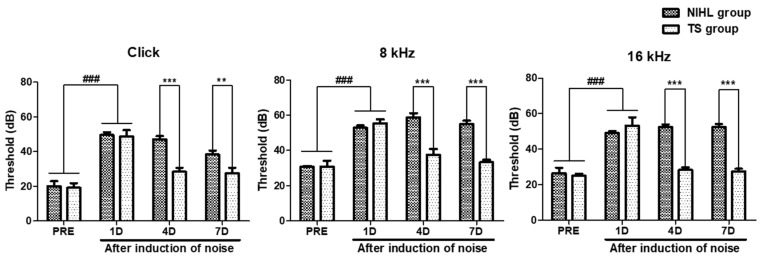
Comparison of hearing thresholds in noise-induced hearing loss mice with and without TS treatment. Hearing thresholds of noise-induced hearing loss (NIHL) and TS groups (TS) were measured using clicks and 8 and 16 kHz tone bursts prior to (PRE) and 1, 4, and 7 days (1D, 4D, and 7D) after noise exposure. Data are presented as means ± SEM. Data were analyzed using two-way ANOVA followed by Bonferroni post hoc test. ### *p* < 0.001 (PRE vs. 1D). ** *p* < 0.01 and *** *p* < 0.001 (NIHL vs. TS). N = 10 per group.

**Figure 5 nutrients-15-01773-f005:**
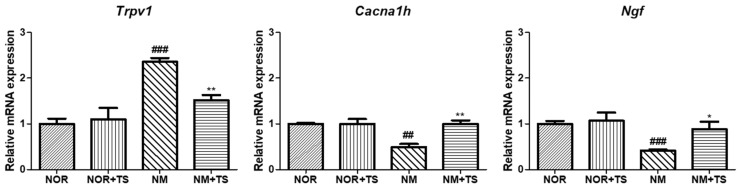
Comparison of gene expression of *Trpv1*, *Ngf*, and *Cacna1h* in neomycin-treated HEI-OC1 Cells. Data are presented as means ± SEM of three independent experiments run in triplicate. ## *p* < 0.01, ### *p* < 0.001 (NOR vs. NM). * *p* < 0.05 ** *p* < 0.01 (NM vs. NM + TS).

**Table 1 nutrients-15-01773-t001:** Primer sequences for RT-PCR.

Gene	Primer	Sequence (5′ to 3′)	NCBI Sequence
*Trpv1*	Forward	GGAAGACAGATAGCCTGAAG	NM_001001445.2
	Reverse	GAGAATGTAGGCCAAGACC	
*Cacna1h*	Forward	GCTCTACTTCATCTCCTTCC	NM_021415.4
	Reverse	CTGTGGCCATCTTCAGTAG	
*Ngf*	Forward	TGAAGCCCACTGGACTAA	NM_001112698.2
	Reverse	GTCTATCCGGATGAACCTC	
*β-actin*	Forward	GAAGAGCTATGAGCTGCCTGA	NM_007393.5
	Reverse	TGATCCACATCTGCTGGAAGG	

## Data Availability

Data is contained within the article.

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
