# Peer review of "Amelioration of Sensorineural Hearing Loss through Regulation of *Trpv1*, *Cacna1h*, and *Ngf* Gene Expression by a Combination of Cuscutae Semen and Rehmanniae Radix Preparata"

_nutrients, 2023, doi:10.3390/nu15071773_

Round 1

Reviewer 1 Report

The manuscript of Hong et al. is well structured and clearly describes the reasonings and the procedures of the various experiments. I still have some questions and comments, though.

Figures 1, 2, 6: why is the number of experiments missing? Figures 1, 2, 3. 6: was TS also significantly different vs. NOR? If this were true, why is it not shown?

Figure 5: are NIHL and TS defined by open and black bars? Since this seems to be the case, it should be explained in the figure legend and the redundant "noise and TS +/- symbols" can be omitted from the figure. This experiment has two variables: TS treatment and time (repeated measurements at different days). Thus, the used statistical procedure One Way ANOVA is not appropriate.

Figure 3: At least as far as I know the used dye YO-PRO-1 selectively stains apoptotic cells, but does not stain healthy cells. Thus, pictures in Fig. 3 suggest that number of apoptotic cells is higher in normal and TS treated larvae, and lower in neomycin-treated larvae. This would be in contrast to authors' claims.

Figure 4: I cannot detect obvious morphological differences in the cilia structure between NOR and NM. The mentioned quotation in the discussion section (#45) does not show any hair cells at all. Scanning electron microscopic pictures of hair cells from humans or mice (see example in quotation #46) show villi in a very regular manner. The pictures shown here do not support authors' claims, at least in my opinion.

I also have difficulties to follow the statements regarding oxidative damage and the role of glutathione in neomycine-induced hair cell loss. In most quotations mentioned by the authors about the protective role of glutathione in hair cell pathology (e.g. #13, 33, 34) neither glutathione nor reactive oxygen species were mentioned. In #14, another aminoglycoside, gentamicin, was used to induce ototoxicity in mice cochleae. In these experiments, the aminoglycoside did not reduce GSH content. Is it possible that authors mistook some relevant publications?

Author Response

Dear Reviewer 1,

Thank you for taking the time to review our paper, "[Amelioration of Sensorineural Hearing Loss through Regulation of Trpv1, Cacna1h, and Ngf Gene Expression by a Combination of Cuscutae Semen and Rehmanniae Radix Preparata]." We greatly appreciate your thoughtful comments and suggestions, which have helped us improve the quality of our work.

Please find attached our response to your review in a separate Word document. We have addressed each of your comments and made the necessary revisions to the manuscript accordingly. We hope that you find our responses satisfactory and that the revised manuscript meets your expectations.

Once again, we would like to express our gratitude for your feedback and constructive criticism. We believe that your insightful comments have made our paper stronger, and we are confident that it will make a valuable contribution to the field.

Thank you for your time and consideration.

Response to Reviewer 1 Comments

  1. Figures 1, 2, 6: why is the number of experiments missing? Figures 1, 2, 3. 6: was TS also significantly different vs. NOR? If this were true, why is it not shown?

: Thank you for this comment. The number of experiments was added in the manuscript. Also, if there is significance between NOR and TS group, we explained the significance on graphs and legends.

Line 236: Data are presented as means ± SEM of three independent experiments in triplicates.

Line 290: Data are presented as means ± SEM of three independent experiments run in triplicate.

Line 235: Figure 1. Comparison of Cell Viability in Neomycin-Treated HEI-OC1 Cells with and without TS Treatment. Data are presented as means ± SEM of three independent experiments in triplicates. ##p < 0.01 (NOR vs. NM+TS). ###p < 0.001 (NOR vs. NM). ***p < 0.001 (NM vs. NM+TS).

Line 245: Figure 2. Comparison of Otic hair cells in Neomycin-Treated Zebrafish with and without TS Treatment. Data are presented as means ± SEM. ###p < 0.001 (NOR vs. NM and NOR vs. NM+TS). **p < 0.01 (NM vs. NM+TS). N = 10 per group.

  1. Figure 5: are NIHL and TS defined by open and black bars? Since this seems to be the case, it should be explained in the figure legend and the redundant "noise and TS +/- symbols" can be omitted from the figure. This experiment has two variables: TS treatment and time (repeated measurements at different days). Thus, the used statistical procedure One Way ANOVA is not appropriate.

: Thank you for this comment. As your advice, I have conducted additional statistical analysis using 2-way ANOVA and have also made modifications to the figure.

Line 216: 2.11. Statistical Analysis Data analysis was performed using the GraphPad Prism software version 5 (GraphPad Software, Inc., San Diego, CA, USA). The results are expressed as the mean ± standard error of the mean (SEM). Statistical significance was evaluated using a one-way analysis of variance (ANOVA) test followed by Tukey's multiple comparison test. Hearing Threshold analysis was performed by two-way ANOVA test followed by Bonferroni post-hoc test. The paired t test was employed to evaluate particular data when needed. A significance level of p < 0.05 was established as the criterion for statistical significance.

Line 271: Figure 4. Comparison of Hearing Threshold in Noise-Induced Hearing Loss Mice with and without TS Treatment. Hearing threshold of noise-induced hearing loss (NIHL) and TS groups (TS) were measured using click and 8, 16 kHz tone burst prior to (PRE), and on 1, 4, and 7 day (1D, 4D, and 7D) after noise exposure. Data are presented as means ± SEM. Data were analyzed using two-way ANOVA followed by Bonferroni post-hoc test. ###p < 0.001 (PRE vs. 1D). **p < 0.01, ***p< 0.001 (NIHL vs. TS). N = 10 per group.

  1. Figure 3: At least as far as I know the used dye YO-PRO-1 selectively stains apoptotic cells, but does not stain healthy cells. Thus, pictures in Fig. 3 suggest that number of apoptotic cells is higher in normal and TS treated larvae, and lower in neomycin-treated larvae. This would be in contrast to authors' claims.

: Thank you for this comment. YO-PRO-1 is a cyanine dye staining the cellular nuclei by binding to DNA. So, it is used to count the number of hair cells in living larvae.

1) MacDonald, G., Raible, D., & Rubel, E. (2002). Zebrafish Neuromast Hair Cell Nuclei are Labeled in Vivo by Uptake of Monomeric Cyanine Dyes. Microscopy and Microanalysis, 8(S02), 1058-1059.

2) Santos F, MacDonald G, Rubel EW, Raible DW. Lateral line hair cell maturation is a determinant of aminoglycoside susceptibility in zebrafish (Danio rerio). Hear Res. 2006;213(1-2):25-33.

  1. Figure 4: I cannot detect obvious morphological differences in the cilia structure between NOR and NM. The mentioned quotation in the discussion section (#45) does not show any hair cells at all. Scanning electron microscopic pictures of hair cells from humans or mice (see example in quotation #46) show villi in a very regular manner. The pictures shown here do not support authors' claims, at least in my opinion.

: Thank you for this comment. In the SEM images, the countable cilia are indicated by red arrows, and it can be seen that the reduced number of cilia in Neomycin-treated group compared to NOR group is recovered by treatment of TS.

Line 254: Figure 3. SEM images of neuromasts in zebrafish larvae. (A) Normal group (NOR) (B) Neomycin-treated group (NM) (C) TS-treated group (NM+TS). Identifiable kinocilia were indicated with red arrows. Scale bar = 1 μm.

  1. I also have difficulties to follow the statements regarding oxidative damage and the role of glutathione in neomycine-induced hair cell loss. In most quotations mentioned by the authors about the protective role of glutathione in hair cell pathology (e.g. #13, 33, 34) neither glutathione nor reactive oxygen species were mentioned. In #14, another aminoglycoside, gentamicin, was used to induce ototoxicity in mice cochleae. In these experiments, the aminoglycoside did not reduce GSH content. Is it possible that authors mistook some relevant publications?

: Thanks for this comment. The data in question has been excluded from the paper after careful consideration, taking into account the advice given by you and the fact that another reviewer mentioned its lack of correlation with the basic logical structure of the paper.

Reviewer 2 Report

The paper sets out to investigate the role of TS, a combination of two natural products in sensorineural hearing loss (SNHL). This is an interesting area of research and there is a clear need to investigate if SNHL can be improved with dietary components. The authors have used both in vivo and in vitro models and this is a strength of the paper. However, there seems to be some disjointed approach to the project. The paper requires a major overhaul and some additional work.

I have the following major:

·         The choice of TS has not been clearly supported by:

o   What does TS stand for?

o   What is the evidence that this particular combination is worth investigating?

o   What is the rationale for 2:1 ratio of Cuscutae Semen and Rehmanniae Radix Preparata

o   How were these two compounds prepared before combining them into this formulation, what form were these two in when they were combined and why?

·         The paragraph 49-54 should not have detail on how the extracts are prepared and focus should be more on ‘why’ these two were chosen?

·         The paragraph 56-66 is not really adding much to the introduction and should be removed.

·         In mice study, how many times were they administered with this particular dose? Is oral route the best? Was it mixed with the normal feed? How much of it was taken in? What about absorption or bioavailability?

·         The introduction lacks adequate rationale to support why these three genes were chosen? TRPV is a logical target to investigate but the justification for NGF and cacna1h is completely missing.

·         Why Cacna1h? why not any other calcium, channel? The genetic variants in the gene that was associated with hearing loss are not even mentioned.

·         If GHS was measured, why not investigate some genes relevant to oxidative stress? This does not fit logically with rest of the study.

·         Viability assay: They have not checked the effect of TS alone on the viability.

·         If TS + NM enhances cell survival, has this been taken into consideration for GSH assay? And RNA isolation and normalisation step at cDNA synthesis?

·         RT-PCR:

o   Why was TS treatment on its own not investigated? It alone could alter the basal level of gene expression.

o   Why was beta actin used as the only endogenous control? There needs to be validation of this gene under experimental conditions and more than one endogenous control needs to be used to support the obtained data.

o   How was it established that the amplification was specific to the genes of interest? Was melting curve analysis conducted?

o   What concentration of RNA was used for cDNA synthesis?

o   The accession number given for beta actin is NOT MOUSE. It is zebrafish.

o   How many cells were used for RNA extraction? it can’t be 96-well plate wells as you could not add 500ul of trizol there? The number of cells is critical as TS seem to increase the viability of cells.

·         Why were ICR mice chosen? Why only males? Why only 6 weeks? This SNHL is age associated, the age is a big factor in this case and is a big omission.

·         Line 95: what is INF-g? Is it interferon gamma? Why would this be used to grow cells?

Minor concerns:

·         Fig 6; The Y-axis should be the same for all three genes.

·         Line 87; spelling of quercetin.

·         Line 53: what do they mean by drying at 9 times?

·         Line 296: dose it mean inner and outer ear air cells?

·         Line 308: induction of neomycin?

·         Line 324: the high quality and accurate genome?

·         A lot of repetition between introduction and discussion

·         Inflammation is a well-researched factor for SNHL and these compounds are anti-inflammatory. It has not been accepted at all.

Author Response

Dear Reviewer 2,

Thank you for taking the time to review our paper, "[Amelioration of Sensorineural Hearing Loss through Regulation of Trpv1, Cacna1h, and Ngf Gene Expression by a Combination of Cuscutae Semen and Rehmanniae Radix Preparata]." We greatly appreciate your thoughtful comments and suggestions, which have helped us improve the quality of our work.

Please find attached our response to your review in a separate Word document. We have addressed each of your comments and made the necessary revisions to the manuscript accordingly. We hope that you find our responses satisfactory and that the revised manuscript meets your expectations.

Once again, we would like to express our gratitude for your feedback and constructive criticism. We believe that your insightful comments have made our paper stronger, and we are confident that it will make a valuable contribution to the field.

Thank you for your time and consideration.

Response to Reviewer 2 Comments

<Major>

  1. The choice of TS has not been clearly supported by:

1-1. What does TS stand for?

: Thank you for this comment. In Korean, Cuscutae Semen is pronounced as “to-sa-ja”, and Rehmanniae Radix Preparata is pronounced as “suk-ji-hwang”. So, the TS was named using the first letter of the words.

1-2. What is the evidence that this particular combination is worth investigating?

: Thank you for this comment. Introduction was revised by referring to your advice.

Line 49: Cuscutae Semen and Rehmanniae Radix Preparata are traditionally used for the purpose of treating or preventing diseases in Korean medicine. Cuscutae Semen refers to the dried ripe seeds of Cuscuta chinensis Lam. [8], and contains various flavonoids such as quercetin and kaempferol [9]. Rehmanniae Radix Preparata refers to the pro-cessed roots of Rehmannia glutinosa Liboschitz ex Steudel, which undergoes a steaming and drying procedure repeated nine times [10]. Rehmanniae Radix is traditionally used in three forms in Korean medicine: raw, dried, and processed. The processing involves nine repetitions of a steaming and drying procedure, which is known to increase the major compound, 5-HMF (5-hydroxymethyl-2-furaldehyde) [11]. Rehmanniae Radix Preparata was reported that has antioxidant effect protecting the damages in mouse auditory cells [12,13]. The combination of Cuscutae Semen and Rehmanniae Radix Preparata has been recorded in traditional Korean medical books such as Donguibo-gam (1613) and Bangyakhappyeon (1884) as being used to treat the symptom of ringing in the ear, which is the perception of sound in the absence of an external sound source. Hearing loss and tinnitus are closely related, and most of the patients with hearing loss are reported to experience tinnitus [14,15]. Therefore, the study was con-ducted to confirm whether the combination of Cuscutae Semen and Rehmanniae Ra-dix Preparata was effective in hearing loss.

1-3. What is the rationale for 2:1 ratio of Cuscutae Semen and Rehmanniae Radix Preparata

: Thank you for this comment. We conducted the in vivo experiment for establish optimal extraction conditions, such as the time and temperature of extraction, the concentration of extraction solvent (EtOH), the ratio of combination, which has the most ameliorative effect in the thresholds of noise-induced hearing loss mice. As the results, the ratio of this combination was determined at 2:1 ratio.

1-4. How were these two compounds prepared before combining them into this formulation, what form were these two in when they were combined and why?

: Thank you for this comment. The Cuscutae Semen means the dried ripe seeds of Cuscuta chinensis Lam., and the Rehmanniae Radix Preparata means the processed roots of Rehmannia glutinosa Liboschitz ex Steudel with nine repetitions of a steaming and drying procedure. So, after mixing the Cuscutae Semen and Rehmanniae Radix Preparata at a ratio of 2:1, extraction and concentration were carried out, and the powdered extract by spray dryer was stored at room temperature. This powdered extract was diluted in an appropriate solvent at the specified concentration before the experiment and used in the experiment. 

  1. The paragraph 49-54 should not have detail on how the extracts are prepared and focus should be more on ‘why’ these two were chosen?

: Thank you for this comment. The paragraph 49-54 was modified to focus on the reasons for choosing these two.

Line 49: Cuscutae Semen and Rehmanniae Radix Preparata are traditionally used for the purpose of treating or preventing diseases in Korean medicine. Cuscutae Semen refers to the dried ripe seeds of Cuscuta chinensis Lam. [8], and contains various flavonoids such as quercetin and kaempferol [9]. Rehmanniae Radix Preparata refers to the pro-cessed roots of Rehmannia glutinosa Liboschitz ex Steudel, which undergoes a steaming and drying procedure repeated nine times [10]. Rehmanniae Radix is traditionally used in three forms in Korean medicine: raw, dried, and processed. The processing involves nine repetitions of a steaming and drying procedure, which is known to increase the major compound, 5-HMF (5-hydroxymethyl-2-furaldehyde) [11]. Rehmanniae Radix Preparata was reported that has antioxidant effect protecting the damages in mouse auditory cells [12,13]. The combination of Cuscutae Semen and Rehmanniae Radix Preparata has been recorded in traditional Korean medical books such as Donguibo-gam (1613) and Bangyakhappyeon (1884) as being used to treat the symptom of ringing in the ear, which is the perception of sound in the absence of an external sound source. Hearing loss and tinnitus are closely related, and most of the patients with hearing loss are reported to experience tinnitus [14,15]. Therefore, the study was con-ducted to confirm whether the combination of Cuscutae Semen and Rehmanniae Ra-dix Preparata was effective in hearing loss.

  1. The paragraph 56-66 is not really adding much to the introduction and should be removed.

: Thank you for this comment. The paragraph 56-66 was removed.

  1. In mice study, how many times were they administered with this particular dose? Is oral route the best? Was it mixed with the normal feed? How much of it was taken in? What about absorption or bioavailability?

: Thank you for this comment. The sentence was changed as following:

Line 170 : Twenty-four hours after noise exposure, the mice in TS group (n=10) were orally ad-ministered 100 mg/kg of TS in distilled water once daily from day 1 to day 7.

  1. The introduction lacks adequate rationale to support why these three genes were chosen? TRPV is a logical target to investigate but the justification for NGF and cacna1h is completely missing.

: Thank you for this comment. Introduction was revised by referring to your advice.

Line 66: Recent studies have linked the genes Trpv1, Cacna1h, and Ngf with sensorineural hearing loss. The Transient Receptor Potential Vanilloid 1 (TRPV1) channel is a cation channel activated by heat and other physical and chemical stimuli [16]. This channel is present in the organ of Corti and mediates cellular uptake of ototoxic aminoglycosides increasing the generation of reactive oxygen species [17,18]. In addition, the activation of TRPV1 by noise and ototoxic drugs in the inner ear leads to depolarization of the membrane and calcium influx, resulting in hearing loss [19-22]. Calcium (Ca2+) is a critical ion involved in regulating various cellular activities. The intracellular homeo-stasis of calcium ions is essential for cell survival and is altered by noise and ototoxic drugs [23-25]. An excessive level of intracellular Ca2+ can activate a series of enzymes that cause cellular damage and apoptosis [26]. The intracellular concentration of calcium is regulated by voltage-gated calcium channels (VGCC), including L, N, P/Q, R, and T-type calcium channels [27]. Among them, the T-type Ca2+ channels encoding by gene of Cacna1h have been reported to play a crucial role in auditory perception and information processing within the inner ear and brainstem [28]. In addition, the TRPV1 activation inhibits the T-type calcium channel in rat sensory neurons [29]. In auditory pathway, spiral ganglion neuron (SGN) is the first afferent neuron [30], and nerve growth factor (NGS) plays a crucial role in the growth, survival, and differentiation of acoustic ganglion cells [31,32]. Loss of cochlear neurons often occurs as a result of hair cell loss, which is thought to be due to decreased neurotrophin exposure and can progress slowly, particularly in humans [33]. Patients with sensorineural hearing loss have been found to have low levels of NGF in their serum [34]. Also, an increase in NGF has been shown to protect against ototoxic drug-induced hair cell damage and improve hearing as measured by the Auditory Brainstem Response (ABR) test [35,36]. Therefore, the study was conducted to confirm the effects of TS on sensorineural hearing loss by regulating the gene expressions of Trpv1, Cacna1h, and Ngf.

  1. Why Cacna1h? why not any other calcium, channel? The genetic variants in the gene that was associated with hearing loss are not even mentioned.

: Thank you for this comment. We did RNA sequencing, and the differentially expressed gene (DEG), cacna1h gene in our data was chosen. Of course, there are reports that other calcium channel-related genes, such as cacna1d are also related to hearing loss [1-3], but RNA sequencing results for the mechanism of TS on sensorineural hearing loss will be posted in a later paper when a significant conclusion is reached.

[1] Liaqat K, Schrauwen I, Raza SI, Lee K, Hussain S, Chakchouk I, Nasir A, Acharya A, Abbe I, Umair M, Ansar M, Ullah I, Shah K; University of Washington Center for Mendelian Genomics; Bamshad MJ, Nickerson DA, Ahmad W, Leal SM. Identification of CACNA1D variants associated with sinoatrial node dysfunction and deafness in additional Pakistani families reveals a clinical significance. J Hum Genet. 2019 Feb;64(2):153-160.

[2] Satheesh SV, Kunert K, Rüttiger L, Zuccotti A, Schönig K, Friauf E, Knipper M, Bartsch D, Nothwang HG. Retrocochlear function of the peripheral deafness gene Cacna1d. Hum Mol Genet. 2012 Sep 1;21(17):3896-909.

[3] Satheesh SV, Kunert K, Rüttiger L, Zuccotti A, Schönig K, Friauf E, Knipper M, Bartsch D, Nothwang HG. Retrocochlear function of the peripheral deafness gene Cacna1d. Hum Mol Genet. 2012 Sep 1;21(17):3896-909.

  1. If GHS was measured, why not investigate some genes relevant to oxidative stress? This does not fit logically with rest of the study.

: Thanks for this comment. As you said, the GSH results do not correlate well with the 3 genes related to hearing loss we selected. Therefore, we will exploit them in future studies after investigating some genes related to oxidative damage caused by neomycin.

  1. Viability assay: They have not checked the effect of TS alone on the viability.

: Thank you for this comment. When only TS was treated in HEI-OC1, the change in cell viability through MTT experiment was as follows, and the data were added.

Line 235: Figure 1. Comparison of Cell Viability in Neomycin-Treated HEI-OC1 Cells with and without TS Treatment. Data are presented as means ± SEM of three independent experiments in triplicates. ##p < 0.01 (NOR vs. NM+TS). ###p < 0.001 (NOR vs. NM). ***p < 0.001 (NM vs. NM+TS).

  1. If TS + NM enhances cell survival, has this been taken into consideration for GSH assay? And RNA isolation and normalisation step at cDNA synthesis?

: Thank you for this comment. We were aware of the possibility that the number of cells used in the initial sample preparation may vary due to the influence of the sample. Thus, when we measure the GSH concentration, one sample was divided into two identical samples, and then the protein concentration was measured using Qubit™ protein assay kit for the correction value between samples.

  1. RT-PCR:

10-1. Why was TS treatment on its own not investigated? It alone could alter the basal level of gene expression.

: Thank you for this comment. When only TS was treated in HEI-OC1, the changes in gene expression in PCR was as follows, and the data were added.

Line 288: Figure 5. Comparison of Gene Expression of Trpv1, Ngf, and Cacna1h in Neomycin-Treated HEI-OC1 Cells. Data are presented as means ± SEM of three independent experiments run in triplicate. ##p < 0.01, ###p < 0.001 (NOR vs. NM). *p < 0.05 **p < 0.01 (NM vs. NM+TS).

10-2. Why was beta actin used as the only endogenous control? There needs to be validation of this gene under experimental conditions and more than one endogenous control needs to be used to support the obtained data.

: Thank you for this comment. To normalize RT-PCR data, the most experiments use the one or more endogenous control genes. We also used several endogenous genes in the early stages of setting up the RT-PCR experiment. However, the beta-actin was stably expressed within the samples to be compared in HEI-OC1 cells, and it can be confirmed that many papers that conducted RT-PCR using HEI-OC1 cells also used only beta-actin as an endogenous gene [1-5].

[1] Yin H, Sun G, Yang Q, Chen C, Qi Q, Wang H, Li J. NLRX1 accelerates cisplatin-induced ototoxity in HEI-OC1 cells via promoting generation of ROS and activation of JNK signaling pathway. Sci Rep. 2017 Mar 13;7:44311.

[2] Xing Y, Ming J, Liu T, Zhang N, Zha D, Lin Y. Decreased Expression of TRPV4 Channels in HEI-OC1 Cells Induced by High Glucose Is Associated with Hearing Impairment. Yonsei Med J. 2018 Nov;59(9):1131-1137.

[3] Gao SS, Choi BM, Chen XY, Zhu RZ, Kim Y, So H, Park R, Sung M, Kim BR. Kaempferol suppresses cisplatin-induced apoptosis via inductions of heme oxygenase-1 and glutamate-cysteine ligase catalytic subunit in HEI-OC1 cell. Pharm Res. 2010 Feb;27(2):235-45.

10-3. How was it established that the amplification was specific to the genes of interest? Was melting curve analysis conducted?

: Thank you for this comment. The specificity of the amplified product was confirmed by conducting melting curve analysis and determining the melting temperature (Tm).

Line 206: The primer sequences used for RT-PCR are listed in Table 1. The RT-PCR was carried out with the following conditions: one cycle at 95°C for 5 minutes, followed by 45 cy-cles of 95°C for 15 seconds, 60°C for 15 seconds, and 72°C for 20 seconds, followed ad-ditional one cycle at 72°C for 20 seconds and then an increase in temperature to 95°C at 0.1°C/s to confirm the specificity of each PCR reaction.

10-4. What concentration of RNA was used for cDNA synthesis?

: Thank you for this comment. The 1 μg of total RNA was used for cDNA synthesis. This information was added in 2.11. Quantitative RT-PCR.

Line 202: Complementary DNA (cDNA) was synthesized using the 1 μg of total RNA and ReverAid First Strand cDNA Synthesis Kit (Thermo Fisher Scientific Korea Ltd., Seoul, South Korea) according to the manufacturer's instructions.

10-5. The accession number given for beta actin is NOT MOUSE. It is zebrafish.

: Thank you for this comment. The accession number was changed correctly.

Line 215: Table 1. Primer sequences for RT-PCR.

Gene

Primer

Sequence (5’ to 3’)

NCBI sequence

Trpv1

Forward

GGAAGACAGATAGCCTGAAG

NM_001001445.2

Reverse

GAGAATGTAGGCCAAGACC

Cacna1h

Forward

GCTCTACTTCATCTCCTTCC

NM_021415.4

Reverse

CTGTGGCCATCTTCAGTAG

Ngf

Forward

TGAAGCCCACTGGACTAA

NM_001112698.2

Reverse

GTCTATCCGGATGAACCTC

β-actin

Forward

GAAGAGCTATGAGCTGCCTGA

NM_007393.5

Reverse

TGATCCACATCTGCTGGAAGG

10-6. How many cells were used for RNA extraction? it can’t be 96-well plate wells as you could not add 500ul of trizol there? The number of cells is critical as TS seem to increase the viability of cells.

: Thank you for this comment. For PCR, the cells were seeded in 6 well cell culture plate at a density of 2 x 106 cells/well. These contents were added in manuscript.

Line 110: The cells were seeded at a density of 1 x 104 cells/well in 96-well flat bottom plates for MTT assay or 2 x 106 cells/well in 6-well cell culture plates for RT-PCR. The incubation was carried out overnight for attachment.

  1. Why were ICR mice chosen? Why only males? Why only 6 weeks? This SNHL is age associated, the age is a big factor in this case and is a big omission.

: Thank you for this comment. Our lab has experience in the validation study of auditory brainstem response (ABR) in ICR mice [1]. Due to the hormonal influence of females, only males were used in experiments. The effects of TS on age-related hearing loss (ARHL) are currently under study.

[1] Hong, B.N.; Park, T.G.; Hong, H.N.; Kang, T.H. A Validation Study of Auditory Brainstem Response (ABR) in ICR Mouse. Audiol Speech Res. 2008, 4, 58-63.

  1. Line 95: what is INF-g? Is it interferon gamma? Why would this be used to grow cells?

: Thank you for this comment. HEI-OC1 cells derive from the auditory organ of the transgenic mouse ImmortomouseTM, which harbors a temperature-sensitive mutant of the SV40 large T antigen gene under the control of an interferon-γ-inducible promoter element. Therefore, the culture of this cell line is characterized by using a medium containing INF-γ.

[1] Kalinec GM, Webster P, Lim DJ, Kalinec F. A cochlear cell line as an in vitro system for drug ototoxicity screening. Audiol Neurootol. 2003;8(4):177-189. doi:10.1159/000071059

<Minor>

  1. Fig 6; The Y-axis should be the same for all three genes.

: Thank you for this comment. The graphs were changed.

Line 288: Figure 5. Comparison of Gene Expression of Trpv1, Ngf, and Cacna1h in Neomycin-Treated HEI-OC1 Cells. Data are presented as means ± SEM of three independent experiments run in triplicate. ##p < 0.01, ###p < 0.001 (NOR vs. NM). *p < 0.05 **p < 0.01 (NM vs. NM+TS).

  1. Line 87; spelling of quercetin.

: Thank you for this comment. The identified ingredient is Quercitrin (Quercetin 3-rhamnoside, C21H20O11).

  1. Line 53: what do they mean by drying at 9 times?

: Thank you for this comment. In Republic of Korea, the Rhemannie Radix is traditionally used in three types, such as the raw, dried, and processed materials. The processing of this includes nine repetitions of a steaming and drying procedure. Increasing the repetitions of a steaming and drying procedure is known to increase 5-HMF (5-hydroxymethyl-2-furaldehyde) in Rehmanniae Radix Preparata.

  1. Line 296: dose it mean inner and outer ear hair cells?

: Thank you for this comment. The cochlea is the part of the inner ear, and the organ of Corti is located in the cochlea. So, it means the inner and outer hair cells in the cochlea of inner ear.

  1. Line 308: induction of neomycin?

: Thank you for this comment. The words of "neomycin-induced" was changed to "neomycin-treated" or “treatment of neomycin” throughout the manuscript.

  1. Line 324: the high quality and accurate genome

: Thank you for this comment. These words were deleted.

Line 323 : The zebrafish genome has been extensively studied and has a homologous gene for 84% of the genes known to be associated with human diseases. [50].

  1. A lot of repetition between introduction and discussion

: Thank you for this comment. The manuscript was revised by referring to your advice.

  1. Inflammation is a well-researched factor for SNHL and these compounds are anti-inflammatory. It has not been accepted at all.

: Thank you for this comment. We think that the combination of Cuscutae Semen and Rehmanniae Radix Preparata has various ingredients, which are active through multiple mechanism including anti-inflammatory related. Currently, we are trying to identify the ingredients in this combination through analysis of HPLC or MS, and figure out the significant mechanisms through analysis of differentially expressed genes by GO and KEGG etc. We believe that these efforts could provide the comprehensive understanding of effect on sensorineural hearing loss by TS.

Round 2

Reviewer 1 Report

-

Author Response

Dear Reviewer 1,

Thank you for taking the time to review our paper, "[Amelioration of Sensorineural Hearing Loss through Regulation of Trpv1, Cacna1h, and Ngf Gene Expression by a Combination of Cuscutae Semen and Rehmanniae Radix Preparata]."